# Hepatic Tumor Microenvironments and Effects on NK Cell Phenotype and Function

**DOI:** 10.3390/ijms20174131

**Published:** 2019-08-24

**Authors:** Julián Piñeiro Fernández, Kimberly A. Luddy, Cathal Harmon, Cliona O’Farrelly

**Affiliations:** 1School of Biochemistry and Immunology, Trinity College Dublin, D02 PN40 Dublin, Ireland; 2Department of Cancer Physiology, H. Lee Moffitt Cancer Center, Tampa, FL 33626, USA; 3Brigham and Women’s Hospital, Harvard Institutes of Medicine, Harvard Medical School, Boston, MA 02138, USA; 4School of Medicine, Trinity College Dublin, D02 PN40 Dublin, Ireland

**Keywords:** liver, NK cells, colorectal cancer, colorectal liver metastasis, tumor microenvironment, exhaustion, hypoxia, lactic acid, adenosine, tryptophan, immune checkpoints, hepatocellular carcinoma, immunometabolism

## Abstract

The liver is a complex organ with critical physiological functions including metabolism, glucose storage, and drug detoxification. Its unique immune profile with large numbers of cytotoxic CD8^+^ T cells and significant innate lymphoid population, including natural killer cells, γδ T cells, MAIT cells, and iNKTcells, suggests an important anti-tumor surveillance role. Despite significant immune surveillance in the liver, in particular large NK cell populations, hepatic cell carcinoma (HCC) is a relatively common outcome of chronic liver infection or inflammation. The liver is also the second most common site of metastatic disease. This discordance suggests immune suppression by the environments of primary and secondary liver cancers. Classic tumor microenvironments (TME) are poorly perfused, leading to accumulation of tumor cell metabolites, diminished O_2_, and decreased nutrient levels, all of which impact immune cell phenotype and function. Here, we focus on changes in the liver microenvironment associated with tumor presence and how they affect NK function and phenotype.

## 1. Introduction

“*The myth of Prometheus means that all the sorrows of the world have their seat in the liver. But it needs a brave man to face so humble a truth.*”—François Mauriac, in Le Nœud de vipères (1932)

From ancient Greeks to modern biologists, the liver has long fascinated humans. Its large, yet unassuming exterior veils the profound functions and unique behavior of this vital organ. The liver is the only internal organ with the capacity to self-regenerate. Once thought to harbor all human emotions, the liver is now known to be the birthplace of fetal blood cells, the key site for detoxification, and the primary location of energy generation and storage. It controls our appetites, mediates inflammation, and modulates immune activity [1,2,3]. Unlike other organs, the liver receives blood from two separate systems: 30% of its supply arrives from the circulation, while the remaining 70% comes from the hepatic portal system that drains from the GI tract, gallbladder, pancreas, and spleen, carrying a multitude of harmless dietary and microbial antigens. Although classified as a non-lymphoid organ, the immunological composition of the liver is strikingly unique, containing large populations of innate lymphocytes with strong anti-tumor potential, including natural killer cells (NK cells), natural killer T cells (NKT cells), mucosal-associated invariant T cells (MAIT cells), and gamma delta T cells (γδ T cells) [4,5,6,7]. These fast-responding cytotoxic cells are charged with protecting the liver and hence the rest of the body from ingested pathogens and transformed hepatocytes, as well as disseminated tumor cells arriving in the hepatic vein.

NK cells, which make up to 50% of the liver lymphocyte population, are cytotoxic cells with anti-tumor functions that are mediated through the release of cytotoxic granules, TRAIL and FasL [5]. Unlike their adaptive counterparts, CD8 T cells, NK cells do not rely on antigen presentation; instead, they are activated through a cascade of various activating and inactivating receptors (Figure 1). This allows NK cells to target stressed and damaged self cells. Liver NK populations include high proportions of CD56^bright^ cells and also a population of liver-resident NK cells, which are characterized by higher expression of CXCR6 and CD69, altered expression of the transcription factors Eomes and Tbet, and exhibit a strong cytotoxic function [2,5,8]. Despite being enriched with large numbers of NK cells, malignant cells can embed and thrive in some livers.

Hepatocellular carcinoma (HCC) accounts for the vast majority of primary liver cancers, which originate from hepatocytes that become transformed. Chronic inflammation caused by infection, autoimmunity, obesity, excess alcohol, and iron accumulation [9,10,11,12,13,14] are primary drivers of hepatocyte transformation. Liver metastases or secondary cancers are typically from primaries of the GI tract, but can also include breast, lung, and melanoma, which often arrive through the portal vein [15]. The liver provides quite a different microenvironment to other organs, perhaps explaining why classic checkpoint inhibition has hitherto failed in patients with liver metastases [16,17]. Changes in nutrient flow, vasculature, and immune profiles create selection pressures for arriving cancer cells. Only those most fit for the new environment will survive and repopulate the new site, changing the local microenvironment in the process. Here, we discuss the conditions of the tumor microenvironment in the liver and its effects on the phenotype and functions of hepatic NK cells.

## 2. Hepatic Tumor Microenvironment

“*Prometheus gave us fire and we use it to light cigarettes.*”—Marty Rubin

The tumor microenvironment (TME) is formed when cancer cells begin to invade and change the surrounding tissue architecture. Composed of tumor cells, immune cells, and stromal cells, the TME is a unique ecosystem providing a protective niche for tumor cells to thrive [6,18]. Common abiotic features of the TME include low oxygen concentrations, altered metabolite accumulations, acidic pH, as well as a collection of immunosuppressive cytokines and growth factors (Figure 2). These harsh conditions not only shape the tumor cell repertoire, but also strongly impact the health and function of resident and tumor-infiltrating immune cell populations. The TME can:
drive differentiation of regulatory immune cells [19,20],inhibit immune cell activation [21,22],induce death or halt proliferation of immune cells [19,23].


NK cells are no exception. The various features of the TME can suppress NK cell function, skew their differentiation, change their phenotype, halt their proliferation, and even induce apoptosis.

### 2.1. Hypoxia

The healthy liver is marked by regions of high and low oxygen levels, fitting the needs of hepatocytes tasked with different metabolic roles [24,25]. While much of the normal liver is highly vascularized, poor vessel perfusion and rapid uptake of O_2_ by densely-packed tumor cells often results in hypoxic environments within the tumor. With a median oxygen level of 0.8%, hepatic cellular carcinoma is one of the most hypoxic tumor types [26]. Furthermore, the perivenous portions of the liver, hypothesized to receive the majority of disseminated tumor cells, have low oxygen tension [15]. Cancer cells arriving from primary sites in the GI tract, the breast, and the lung must undergo metabolic changes through the regulation of hypoxia-induced transcription factors (e.g., HIF-1α and c-Myc), which will help develop low-oxygen tolerance to survive this hypoxic environment [27]. Highly-glycolytic cancer cells express HIF-1α, which under normal conditions, is ubiquitinated and degraded in the cytoplasm [28]. However, when oxygen is low, HIF-1α re-enters the nucleus and binds Hif-1β, stabilizing it as a protein and exerting its function as a transcription factor [29,30]. In the liver, hypoxia-derived HIF-1α induces changes in surface and soluble MHC class I polypeptide-related sequence A (MICA), thus impairing NK cell’s ability to recognize the tumor [23,31]. In some cases, Hif-1α, along with other hypoxia-related pathways such as unfolded protein response, also drive tumor cell autophagy [32]. Reversal of autophagy by silencing important genes involved in this process, such as *Beclin1*, increases NK cell infiltration into the tumor [33].

Hypoxia can have diverse direct effects on NK cells and may even be beneficial for their function under short-term exposure [34,35]. However, following long-term oxygen deprivation, NK cells themselves upregulate HIF-1α [36], resulting in an altered transcriptional profile [34]. Hif-1α downregulates the expression of natural cytotoxicity receptors, NKp30, NKp44, NKp46, and the natural killer group 2D (NKG2D) receptor, activators of NK cells [36]. HIF-1α regulates important genes related to metabolism, cell proliferation, and apoptosis. Metabolic effects of Hif-1α on NK cells include the altered expression of glycolytic enzymes (e.g., PMK2 and PGK1) [37], metabolite transporters, (e.g., GLUT1 and 3, SLC1A5, and MCT4) [37], and enzymes involved in biosynthesis (e.g., FAS and 6PGDH) [38]. Hypoxia inactivates mammalian target of rapamycin (mTOR) in NK cells [39], a protein complex that senses nutrient deficits and controls NK cell growth, maturation, and differentiation [40]. The mechanism is not entirely defined, but it is clear that HIF-1α activation leads to DNA damage and replication arrest, which inhibits mTOR through regulation of DNA damage response 1 (REDD1) [41]. It may also promote degradation of granzyme B through autophagy, as occurs during starvation [42]. Inhibition of mTOR signaling in hepatic NK cells by inactivating or blocking the mTORC1 pathway (gene knockout) also results in the reduction of mature NK cells (lower numbers of CD11b+ cells) and loss of IFNγ production downstream of NKG2D activation and impaired OXPHOS metabolism [43], showing the importance of this pathway in hypoxia-related processes. Hypoxic conditions also reduce intracellular granzyme B and perforin [44].

The acquisition of new blood vessels alleviates the hypoxic burden on tumor cells, allowing for uncontrolled growth. While NK cells are the primary effector cells of the innate immune system, there are subsets of NK cells with differing phenotypes. Decidual NK cells are highly angiogenic cells with a pivotal role in pregnancy [45,46]. Diminished oxygen levels and increased TGFβ in the TME can polarize NK cell differentiation into a proangiogenic phenotype [46,47,48]. Proangiogenic genes, vascular endothelial growth factor (VEGF) and TGF-β, are upregulated in immune cells, usually in a Hif-1α-dependent manner [37]. Particularly, tumor-infiltrating NK cells are more likely to develop a CD56 ^bright^ phenotype [49] upon interaction with PD-L1 in hypoxia [49]. These CD56 ^bright^ NK cells express lower levels of STAT5, necessary for NK cell immunosurveillance [50]. The lack of STAT5A and STAT5B promotes higher levels of the immunosuppressive cytokine TGFβ1, which upregulates VEGF in healthy NK cells (along with others like P1GF) [45]. VEGF binds VEGF receptor 2 (VEGFR2) on the surface of endothelial cells, inducing the formation of new blood vessels. In liver cancers, agrin expression induces endothelial cell recruitment and adhesion to the tumor site, thus promoting angiogenesis and upregulating and stabilizing VEGFR2 [51].

### 2.2. Tumor Metabolism

Cell metabolism undergoes drastic changes during transformation (Figure 3). Under normal conditions, epithelial cells, as well as hepatocytes generate ATP from glucose slowly and efficiently by oxidative phosphorylation [52]. However, cancer cells switch from oxidative phosphorylation (OXPHOS) to aerobic glycolysis as a primary means of glucose metabolism, to meet the increased energy and biomolecular needs of transformation [53,54,55,56,57,58,59,60]. Production of ATP is far less efficient during glycolysis, with just two molecules of ATP generated for each molecule of glucose in contrast to the 37–39 produced via OXPHOS. However, aerobic glycolysis skips the OXPHOS pathway and quickly generates ATP along with a wealth of biomolecules involved in the anabolism and biosynthesis of new structures for cell growth and proliferation [61,62,63,64,65]. One of the most important of these is the pentose-phosphate pathway (PPP), which transforms glucose-6-phosphate, an intermediary metabolite from glycolysis, into ribulose-5-phosphate, which will eventually become de novo-synthesized nucleotides. NADPH generated in the process will be used in lipid synthesis [64,65]. As previously reviewed by Gillies et al., the glycolytic shift in cancer cells involves a wide variety of pathways including hypoxia-inducible factor, myc, PI3K/Akt/mTOR, p53, and Ras [66]. The dramatic increase in glucose metabolism leads to an equally dramatic increase in metabolic end products, which accumulate in the tumor microenvironment and elicits immunoregulatory functions on intratumoral NK cells.

#### 2.2.1. Lactate

Glycolysis generates several intermediate products, which contribute to biosynthesis pathways or NADH production (see [67] for more details). One key intermediate, pyruvate, the conjugate base of pyruvic acid, can be transported into the mitochondria for use in the citric acid cycle, converted to fatty acids, or carbohydrates through gluconeogenesis. In oxygen-poor environments and transformed cells, a large portion of pyruvate is converted to lactate by lactate dehydrogenase [67]. When the intracellular levels of lactate become too high, the proton-linked monocarboxylate transporters (MCT) pump lactate outside of the cell.

The healthy liver is the primary site of lactate absorption via gluconeogenesis [68]. However, in advanced liver disease and cancer, lactate production is increased, and lactate clearance is decreased. This may lead to a build up of lactate in the liver microenvironment [69,70]. Previous work from our group demonstrated that conditioned media derived from ex vivo cultures of colorectal cancer liver metastasis tissues had elevated levels of lactate (>10 mmol/L) compared to non-cancer liver tissue controls (<3 mmol/L) [69]. High tumor lactate levels are a poor prognostic indicator in many cancers including liver cancers [69]. Hepatic cytotoxic lymphocytes, such as NK cells and CD8 T cells, become inactivated when levels of intracellular lactate are high, either by downregulating the expression of NKp46, CD107a, and granzyme B [71] or simply by undergoing apoptosis due to the mitochondrial damage induced by reactive oxygen species, particularly in the hepatic CD56^bright^ subpopulations [69].

#### 2.2.2. Adenosine

Adenosine is a purine metabolite that suppresses NK cell proliferation, maturation [72], and effector function through the mTOR pathway [73]. Adenosine is generated by tumor cell cleavage of ATP into AMP and then AMP into free adenosine (the responsible enzymes for these processes are the ectonucleotidases CD39 and CD73, respectively) [74]. Under low oxygen concentrations within the tumor, neoplastic cells and some Tregs [75] express higher levels of CD39 and CD73 in an HIF-1α/mTOR-dependent manner [76]. Additionally, NK cells can synthesize and secrete adenosine themselves. CD56^bright^ NK cells produce free adenosine via CD38 and CD203a, acting as an immunosuppressor cell by regulating other lymphocytes, mainly CD4+ T cells, while CD56^dim^ express lower levels of CD73 and CD39 [77].

This metabolite not only suppresses NK cell function, but many other immune cells including T cells, and recruits and activates other immunosuppressive cells such as regulatory T cells [78] and myeloid-derived suppressor cells (MDSCs). MDSC inhibit CD4 helper T cells, a primary source of NK cell-supportive cytokines. All of this decreases the activity and the killing potential of NK cells, but it does not directly affect the antibody-dependent cell cytotoxicity (ADCC) of NK cells [23], although antibody secretion is impaired due to the effects of the tumor on antibody-producing plasma cells [79,80]. Blocking adenosine signaling reverses the immunosuppressive effects and may be a potential therapeutic option in combination with other NK cell-based immunotherapies, including adoptive transfer and checkpoint blockade [39,81,82].

#### 2.2.3. Tryptophan Catabolism

Tryptophan is an essential amino acid catabolized through either the serotonin or kynurenine pathways. In the former, tryptophan is used by the nervous system to produce serotonin and is required for the synthesis of melatonin and vitamin B3 [83]. The kynurenine pathway is mediated by intracellular enzymes, indoleamine-2,3-dioxygenase (IDO) and tryptophan-2,3-dioxygenase 2 (TDO). Kynurenine is further processed to quinolinic acid and picolinic acid or released from the cell via CD98 [84]. IDO is an IFN-γ-inducible gene expressed in the thymus, GI tract, lung, maternal–fetal interface, and importantly, by regulatory myeloid cells and antigen-presenting cells [85]. TDO expression is induced by glucocorticoids in hepatic cells and is responsible for maintaining systemic tryptophan levels [85,86,87]. Both enzymes are overexpressed in various cancer types. Constitutive and inducible IDO is expressed at high levels in colorectal cancer and is significantly correlated with liver metastasis [88]. TDO is constitutively expressed in glioma, bladder cancer, melanoma, and HCC [86].

The tryptophan/kynurenine pathway is associated with immune suppression in cancer. Overexpression of IDO and TDO by cancer cells, tumor-infiltrating myeloid cells, and aberrant hepatocytes causes a depletion of tryptophan in the surrounding environment. Munn et al. and others hypothesized that tryptophan depletion is responsible for the immunosuppressive effects of tumoral IDO [85]. They showed that without adequate tryptophan levels, lymphocytes are unable to proliferate [89]. Others hypothesized that downstream metabolites including l-kynurenine, 3-hydroxykynurenine, 3-hydroxyanthranilic acid, and picolinic acid are toxic to lymphocytes. At high levels, these metabolites inhibit NK cell proliferation by inducing cell cycle arrest [90]. l-kynurenine blocks cytokine upregulation of NK cell-activating receptors NKp46 and NKG2D at both the transcript and protein level [91]. Furthermore, l-kynurenine inhibited the cytotoxic effects of activated NK cells. These effects are reversible upon removal of the metabolite [91]. In hepatocellular carcinoma, tumor-associated fibroblasts suppress hepatic NK cell activity through IDO and prostaglandin E2 (PGE2, an immunosuppressive lipid product of arachidonic acid metabolism) secretion [92] and can be reversed or palliated by anti-IDO [93]. The mechanisms driving the inhibition of NK cells by either tryptophan depletion or metabolite accumulation are unknown, but may be guided by nutrient sensing.

### 2.3. Acidic pH

Tumor cells consume and breakdown large amounts of glucose through aerobic and anaerobic processes. The outcome is the secretion of acids in the form of lactate and CO_2_ [66]. Carbonic anhydrases (CA) convert CO_2_ into bicarbonate and protons, resulting in acidification of the local environment. CA-IX is a membrane-bound and soluble CA. It is expressed in response to hypoxia and acidosis and correlates with poor prognosis in cirrhosis and HCC [94,95]. Lactate, the secreted end product of glycolysis, further acidifies the environment when hydrogens are removed by ion transporters [66]. In the TME, this build-up of protons can induce a drop in the extracellular pH to 6.6, causing dramatic effects on infiltrating immune cells [96].

The acidification of the tumor microenvironment has different effects on immune cells. On the one hand, it has been shown to increase the maturation of dendritic cells, as well as the activation and recruitment of MDSCs and Treg cells [97], while driving M2 differentiation [98,99,100]. On the other hand, low pH inhibits cytotoxic functions of lymphocytes and blocks secretion of inflammatory cytokines such as IFNγ [101] and TNFα [96]. This effect is amplified in the spleen compared to liver-resident NK cells in rat models, suggesting some acid adaptation of liver-resident NK cells, as they exhibit lower rates of cell death, better morphology, and higher accumulation of granules compared to splenic NK cells in the same acidic environment [102]. Systemic buffering in murine models restored IFNγ expression by NK cells [101] and inhibited the formation of hepatic metastasis [103].

Interestingly, when tumor-related acidosis is extrapolated to other diseases and other microenvironments, such as the microenvironment generated by cryptococcoma (*Cryptococcus gattii* and *Cryptococcus neoformans* infections), the acidic pH in the center of the mass (analogous to the tumor mass) produces increased NK cell degranulation and perforin-mediated killing compared to pH 7.4 [104]. This suggests that acidic pH on its own may not be the sole inducer of the diminished anti-tumor activity of NK cells. Rather, it may be that low pH boosts the immunosuppressive effects of other components of the TME.

### 2.4. Cytokines and Growth Factors

The healthy liver is rich in a plethora of cytokines (e.g., IL-1α, IL-1β, TNFα, and IL-6) and growth factors (e.g., IGF-I, IGF-II, EGF, and TGF-α) that stimulate hepatocyte growth and induce activation upon the correct stimuli [105]. Following tissue injury, Kupffer cells, liver-specific macrophages, secrete pro-inflammatory cytokines such as IL-1, IL-6, and TNF-α [106,107]. TNF-α plays a key role in maintaining liver homeostasis by promoting tissue regeneration along with IL-6 in an STAT3/NFκB-dependent manner [108]. As the wound is repaired, Kupffer cells undergo a shift in phenotype and secrete anti-inflammatory cytokines such as IL-4 and IL-10 [109]. Persistent liver injury and chronic liver disease create a state of perpetual inflammation that can support malignant transformation [110]. Pro-inflammatory cytokine levels are particularly high at early stages of tumor development, indicating that during tumor formation, a pro-inflammatory environment plays a role in tumor development [111]. High levels of pro-inflammatory cytokines such as IL-6, IL-8, and TNF-α are present in the tissue from patients undergoing resection for CRLM [112] and are associated with poor prognosis and tissue damage [113].

While the cytokines and growth factors associated with inflammation may support tumor growth at these early stages, they also activate cytotoxic immune cells, strengthening a cycle of immune-mediated tumor cell death and birth. The term “immunoediting” is often used to describe an evolutionary process on tumor development driven by the immune system [114,115,116]. For example, IL-21, secreted mainly by helper follicular T cells, NKT cells, and Th17 cells, increases the cytotoxicity and proliferation of NK cells and other immune cells [117,118]. It has been shown that IL-21 and IL-2 have synergistic effects, and when combined, the expression of CD25, NKG2A, and CD86 on CD56^bright^ NK cells is upregulated [119]. NKp44 and NKG2D are only upregulated in the presence of IL-2, but with IL-21 costimulation, NKG2D is slightly downregulated, suggesting a negative effect of IL-21 on NKG2D expression [120]. IL-15 promotes NK cell activation, proliferation, and survival [121] via PI3K, PDPK1, and AKT [39], and its expression is tissue-specific. IL-15 is secreted by monocytes, tissue-resident Kupffer cells, activated macrophages, and epithelial cells [122,123] and plays a role in antiviral responses in the liver. IL-15, which shares a similar structure with IL-2, also plays a key role in overcoming NK cell dysfunction and exhaustion by restoring the cytotoxic activity in CRLM and HCC [124]. Other cytokines such as IL-13 (and less importantly, IL-33 and IL-5), Th2 type cytokines, are correlated with fibrosis progression in chronic liver inflammation diseases in mice [113], activating pro-fibrotic genes through the STAT6-ZEB1 pathway [125,126].

As the tumor progresses, the cytokine pattern shifts towards an immune-evasive anti-inflammatory response, resulting in diminished activation of the immune system. Upregulated anti-inflammatory cytokines within the TME include IL-10 and TGF-β [127], similar to those observed in chronic inflammation processes [128]. The reasons behind these changes over time are not completely understood, but the physicochemical properties of the tumors are thought to be the main reason [128]. IL-10 induces dysregulation of NK surface receptor expression, including NKG2A. Hepatic NK cells have been shown to upregulate NKG2A and downregulate Ly49 (an MHC I-binding receptor) in the presence of IL-10, resulting in an exhaustion-like phenotype and an inability to respond appropriately to stimuli [129,130]. TGF-β inhibits NK cell function by inhibiting cytokine-mediated activation of mTOR [131]. Furthermore, inhibition of TGF-β signaling restores NK cell functionality in murine models of colorectal cancer liver metastasis [132].

Hepatocellular carcinoma often results from chronic viral infections including, hepatitis C virus, and hepatitis B virus (HCV and HBV). The presence of virus in the tumor can further complicate the immune response [133]. Tumoral HCV and HBV infections have differential effects on the resident NK cell population. High viral loads in patients with established HCC correlate with tumor progression and poor outcomes [134]. Virally-infected cells stimulate monocytes to express IL-18, which activates NK cells, as well as immunosuppressive cytokines, TGFβ, and IL-10 [135]. During the course of viral-induced HCC, hepatic NK cells are exposed to both transformed cells and infected cells. Infected hepatocytes activate NK cells, which can lead to viral clearance. However, chronic infections are often accompanied by diminished NK cell function and IFNγ production [135]. Qa-1, expressed by infected hepatocytes, binds to NKG2A on NK cells and induces exhaustion [136]. Chronic HCV promotes CD56 ^neg^ NK cells with diminished cytotoxicity [137].

## 3. Conclusions

NK cell activation is a complex process that is sensitive to the heterogeneous microenvironments within the tumor (Figure 4). While here, we review the common abiotic components of the liver TME (hypoxia, metabolites, acidosis, cytokines, and growth factors) and their effects on hepatic NK cells, there are many factors beyond the scope of this review that also contribute to reduced intratumoral NK cell function, including:
the biotic players of the TME including stromal cells, regulatory immune cells, and normal neighboring cells that contribute to NK cell inhibition [23,138]the effects of reduced or altered chemokines on NK cells homing to the tumor site [138,139]the potential role of regulatory NK cells in the tumor microenvironment [82,140].


Immunotherapies, particularly checkpoint inhibitors, have revolutionized the way we treat cancers. The immune system now sits at the forefront of drug development and trial design in oncology. However, the liver remains the least responsive organ to current checkpoint immunotherapies, perhaps due to additional checkpoints expressed by NK cells yet to be targeted, as well as the strong immunosuppression of the metastatic liver environment. While T cells are currently a prime target of cancer immunologists, many are beginning to see the potential of NK cells. Their lack of dependence on antigen expression and rapid activation make them an attractive addition to T cell-based immunotherapies, particularly in the prevention of metastatic disease. To further the development of NK cell-based immunotherapy, we must understand the complex and often apparently redundant pathways of activation and inactivation of these highly-cytotoxic cells. The high frequency of liver-resident NK cells in healthy liver suggests an essential role for targeting and preventing liver metastasis.

Organ-specific immune responses, especially tumor surveillance mechanisms, are as diverse as the tissues in which they are housed. This is particularly true in common sites of metastasis: the lymph nodes, liver, and bone marrow. Disseminating cancer cells are faced with novel immunological challenges upon entering new tissue that select only for those tumor cells able to thrive in the face of these new predators. Perhaps the resulting immunologic bottlenecks can be better exploited with organ-specific immunotherapies designed to target the tissue housing the current lesion rather than that of the primary site.

## Figures and Tables

**Figure 1 ijms-20-04131-f001:**
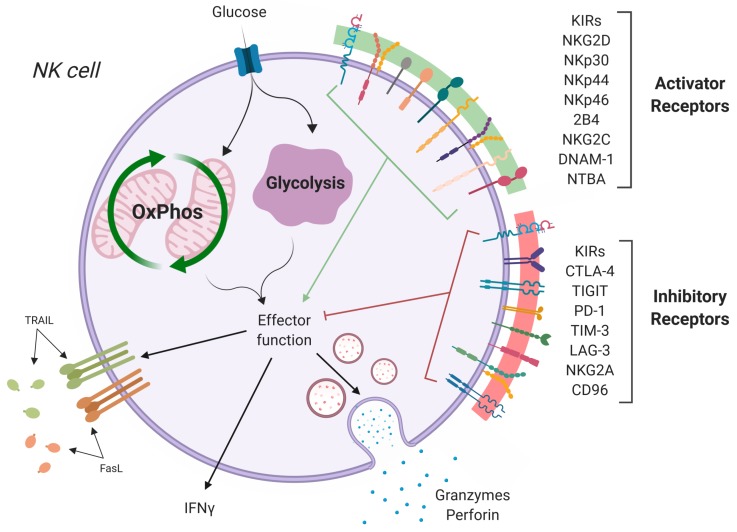
NK cell activation/inhibition. NK cells become activated through a complex network of activating receptors (green) and inhibitory receptors (red). Loss of inhibition or amplification of activating signals trigger NK cell activation, inducing metabolic changes and driving effector functions, including release of cytotoxic granules, pro-inflammatory cytokines (IFNγ), and death receptor signaling (TRAIL, FasL). Figure created with BioRender.com.

**Figure 2 ijms-20-04131-f002:**
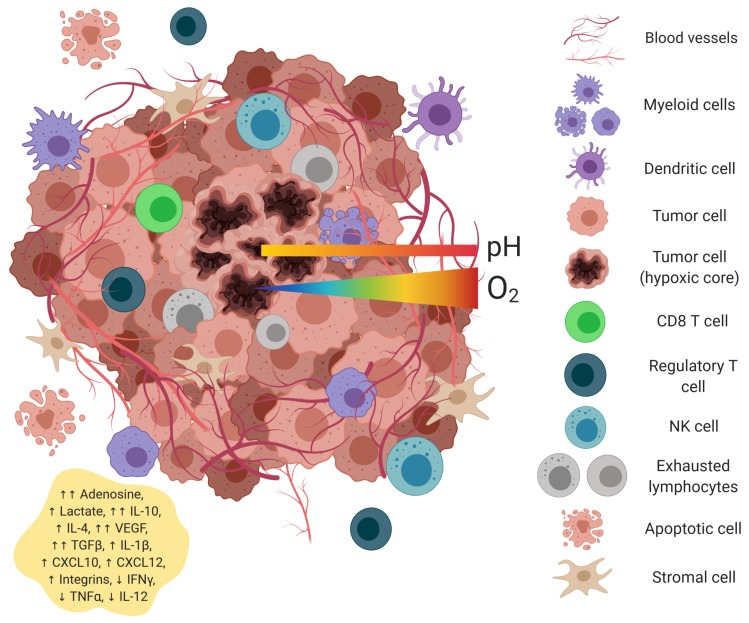
Components of the tumor microenvironment. The tumor microenvironment is a complex ecosystem of heterogeneous tumor cells, stromal cells, and a variety of immune cells residing in a network of dysregulated vasculature and collagen. Poor perfusion and densely-packed glycolytic tumor cells create pockets of diminished oxygen levels, acidic pH, poor nutrient loads, anti-inflammatory cytokines, chemokines, and accumulated metabolic by-products, such as lactate. Tumor-infiltrating immune cells of both the myeloid and lymphoid lineages are found within the TME. Figure created with BioRender.com.

**Figure 3 ijms-20-04131-f003:**
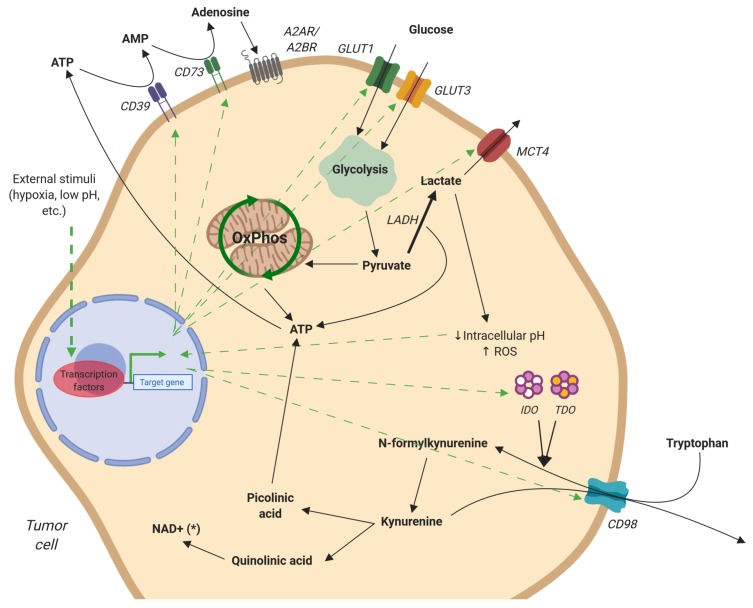
Metabolism of adenosine, glucose, and tryptophan by tumor cells. Free adenosine is produced by the cleavage of extracellular ATP by CD39 and CD73. Signaling through A2AR, adenosine initiates an anti-inflammatory response. Glucose uptake is increased, via GLUT1/3, after which it is converted to pyruvate via glycolysis. Excess pyruvate is transformed into lactate and either exported by MCT4 or remains in the cytoplasm, decreasing cellular pH, resulting in ROS production. Tryptophan, transported by CD98, is transformed to kynurenine by IDO and TDO. Kynurenine is either released from the cell via CD98 or further processed to quinolinic acid and picolinic acid. These intermediates can be used for the generation of NAD+ and ATP, respectively. Green arrows represent stimulation or upregulation. Figure created with BioRender.com.

**Figure 4 ijms-20-04131-f004:**
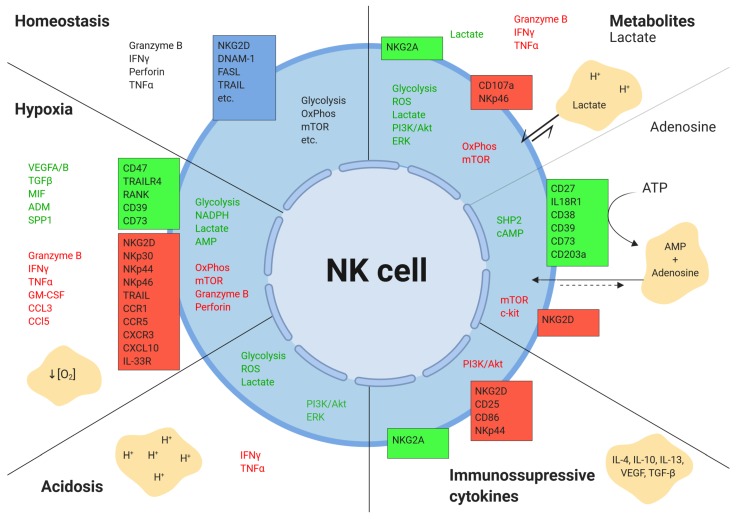
Effects of the tumor microenvironment on NK cells.The tumor microenvironment alters the expression of various secreted, surface, and intracellular molecules in NK cells. Boxes represent surface receptors. Green indicates upregulation, and red depicts downregulation. Figure created with BioRender.com.

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
