# Peer review of "Hepatic Tumor Microenvironments and Effects on NK Cell Phenotype and Function"

_ijms, 2019, doi:10.3390/ijms20174131_

Round 1

Reviewer 1 Report

Authors described hepatic tumor microenvironments and their effects of NK cells.

The manuscript seemed to be well-documented and all figures seemed to be very nice.

The following issues would’ve modified by authors.

1) Regarding Fig. 1:  Fas L and TRAIL looked to be bound on cell surface. However, these are the molecules as soluble factor to act with Fas-receptors or Trail receptors expressed on tumor cells as well as other cells.  Thus, it would be better to modify to understand easily these factors are soluble.

2) Fig 2:  right upper part, there is cells painted in violet. It seems to be dendritic cell, however, there is no description.

3) I think most of HCC are caused by hepatic B or C virus. Of course, as authors described in text, other causes are also concerned to carcinogenesis of hepatic cells. Again, however, majority is HBV or HCV.  Are there any interaction between NK cells and these viruses? If there are, please describe some (even in short description).

Author Response

Dear Reviewer,

Thank you for taking the time and effort to review our paper, ‘Hepatic Tumor Microenvironments and Effects on NK Cell Phenotype and Function’.  We have considered your comments carefully and have undertaken greater work as well as made a few clarifications in the revised manuscript. We genuinely feel that your suggestions have improved the paper. Please see our responses below.

 1) Regarding Fig. 1:  Fas L and TRAIL looked to be bound on cell surface. However, these are the molecules as soluble factor to act with Fas-receptors or Trail receptors expressed on tumor cells as well as other cells.  Thus, it would be better to modify to understand easily these factors are soluble.

We have modified the figure to reflect the soluble potential of FasL and TRAIL. Please see revised figure 1.

2) Fig 2:  right upper part, there is cells painted in violet. It seems to be dendritic cell, however, there is no description.

Description for the dendritic cells has been added. Please see the revised figure 2.

3) I think most of HCC are caused by hepatic B or C virus. Of course, as authors described in text, other causes are also concerned to carcinogenesis of hepatic cells. Again, however, majority is HBV or HCV.  Are there any interaction between NK cells and these viruses? If there are, please describe some (even in short description).

This is a valid and rather interesting point. We have included a brief review of the interactions between the virus and NK cells. The following excerpt has been placed into the final section (lines 275-285):

Hepatocellular carcinoma often results from chronic viral infections including, hepatitis C virus and hepatitis B virus (HCV and HBV). The presence of virus in the tumor can further complicate the immune response (134). Tumoral HCV and HBV infections have differential effects on the resident NK cell population. High viral loads in patients with established HCC correlates with tumor progression and poor outcomes (135). Virally infected cells stimulate monocytes to express IL-18, which activates NK cells, as well as, immunosuppressive cytokines, TGF-beta and IL-10 (136). During the course of viral induced HCC, hepatic NK cells are exposed to both transformed cells and infected cells. Infected hepatocytes do activate NK cells which can lead to viral clearance. However, chronic infections are often accompanied by diminished NK cell function and IFN-gamma production (136). Qa-1, expressed by infected hepatocytes binds to NKG2A on NK cells and induces exhaustion (13). Chronic HCV promotes CD56neg NK cells with diminished cytotoxicity (138).

Once again, we thank the reviewers and the editors for the time and attention to our work.

Respectfully yours,

Kimberly A Luddy

Julian Piñeiro Fernández

Cathal Harmon

And

Cliona O’Farrelly

Reviewer 2 Report

The review by Julián Piñeiro Fernández et al, is interesting, well written and provide a rather comprehensive overview on the alteration of the liver microenvironment during tumor development and possible mechanisms associated to low or deregulated NK cells. The liver is a key vital organ with unique immune profile with large numbers of cytolytic CD8+ T cells and significant innate lymphoid population, including NK cells, which account for up to 50% of the immune cells residing in the liver. The article focuses on the characteristics of liver TME and its negative effects on the phenotype and anti-tumor function of NK cells. These features comprise hypoxia, altered metabolite accumulations, lactate, adenosine, Tryptophan catabolism, acidic pH, and different immunosuppressive cytokines and growth factors.

The figures are well structured and clearly and comprehensively show the explanations presented in the text.

The review exposes satisfactorily the different important components of the altered NK cell functions in an exact and clear way. However, in chapter 2 when the authors illustrate the major effects of TME in 3 points, there is a fourth very important factor, which is the phenomenon of tumor angiogenesis, which allows the tumor to grow uncontrollably, an in particular driven-angiogenesis by innate immune cells, including NK cells.

The authors should therefore include this part and indicate in a general way this mechanism, and in particular in the section on hypoxia, in fact, hypoxia, is a key driven factor to the vascularization of tumors (angiogenic process), and NK cells are very sensitive to oxygen deprivation and this resulted in their altered phenotype and functions.

In this regard it would be useful to cite some important articles regarding this important chapter of tumor growth and tumor evasion, such as:

K. Njah et al. A Role of Agrin in Maintaining the Stability of Vascular Endothelial Growth Factor Receptor-2 during Tumor Angiogenesis. Cell Rep. 2019 Jul 23;28(4):949-965.e7.

A. Albini et al. Contribution to Tumor Angiogenesis From Innate Immune Cells Within the Tumor Microenvironment: Implications for Immunotherapy. Front Immunol. 2018 Apr 5;9:527.

M. De Palma et al. Microenvironmental regulation of tumour angiogenesis. Nat Rev Cancer. 2017 Aug;17(8):457-474.

B. Bassani et al. Natural Killer Cells as Key Players of Tumor Progression and Angiogenesis: Old and Novel Tools to Divert Their Pro-Tumor Activities into Potent Anti-Tumor Effects. Cancers (Basel). 2019 Apr 1;11(4). pii: E461.

And with regard to the TGF-β, and hypoxia effects on NK cells, also:

A. Bruno et al. The proangiogenic phenotype of natural killer cells in patients with non-small cell lung cancer. Neoplasia. 2013 Feb;15(2):133-42.

D.S. Allan et al. TGF-β affects development and differentiation of human natural killer cell subsets. Eur J Immunol. 2010 Aug;40(8):2289-95.

D.B. Keskin et al. TGFβ promotes conversion of CD16+ peripheral blood NK cells into CD16- NK cells with similarities to decidual NK cells. Proceedings of the National Academy of Sciences, vol. 104, no. 9, pp. 3378–3383, 2007.

A.S. Cerdeira et al. Conversion of peripheral blood NK cells to a decidual NK-like phenotype by a cocktail of defined factors. Journal of Immunology, vol. 190, no. 8, pp. 3939–3948, 2013.

Author Response

Dear Reviewer,

Thank you for taking the time and effort to review our paper, ‘Hepatic Tumor Microenvironments and Effects on NK Cell Phenotype and Function’.  We genuinely feel that your suggestions have improved the paper, particularly your guidance to include the pro-angiogenic phenotype of some tumor resident NK cells. We have revised the manuscript to include this topic and the references recommended. Please see our responses in blue below.

The review exposes satisfactorily the different important components of the altered NK cell functions in an exact and clear way. However, in chapter 2 when the authors illustrate the major effects of TME in 3 points, there is a fourth very important factor, which is the phenomenon of tumor angiogenesis, which allows the tumor to grow uncontrollably, an in particular driven-angiogenesis by innate immune cells, including NK cells. The authors should therefore include this part and indicate in a general way this mechanism, and in particular in the section on hypoxia, in fact, hypoxia, is a key driven factor to the vascularization of tumors (angiogenic process), and NK cells are very sensitive to oxygen deprivation and this resulted in their altered phenotype and functions.

We agree that the proangiogenic role of NK cells should be included in this review. To that end, we have reviewed the literature and included the below excerpt in the hypoxia section of the revised manuscript (lines 106-119):

The acquisition of new blood vessels alleviates the hypoxic burden on tumor cells allowing for uncontrolled growth. While NK cells are the primary effector cells of the innate immune system there are subsets of NK cells with differing phenotypes. Decidual NK cells are highly angiogenic cells with a pivotal role in pregnancy (45,46). Diminished oxygen levels and increased TGF-beta in the TME can polarize NK cell differentiation into a proangiogenic phenotype (46-48). Proangiogenic genes, vascular endothelial growth factor (VEGF) and TGF-beta are upregulated in immune cells, usually in a Hif-1alpha dependent manner (37).  Particularly, tumor-infiltrating NK cells are more likely to develop a CD56bright phenotype (49) upon interaction with PD-L1 in hypoxia (49). These CD56bright NK cells express lower levels of STAT5, necessary for NK cell immunosurveillance (50). The lack of STAT5A and STAT5B promotes higher levels of the immunosuppressive cytokine TGF-beta, which upregulates VEGF in healthy NK cells (along with others like P1GF0) (45). VEGF binds VEGF receptor 2 (VEGFR2) on the surface of endothelial cells, inducing the formation of new blood vessels. In liver cancers, agrin expression induces endothelial cell recruitment and adhesion to the tumor site, thus promoting angiogenesis and upregulating and stabilizing VEGFR2 (51).

Once again, we thank the reviewers and the editors for the time and attention to our work.

Respectfully yours,

Kimberly A Luddy

Julian Piñeiro Fernández

Cathal Harmon

And

Cliona O’Farrelly

Round 2

Reviewer 2 Report

The review by Julián Piñeiro Fernández et al. has been improved according to review's requests. Indeed, the authors has included, as suggested, the part explaining the tumor angiogenic process and NK involvement in a clear and complete manner.